# Inhibitory Effect and Mechanism of Dryocrassin ABBA Against *Fusarium oxysporum*

**DOI:** 10.3390/ijms26041573

**Published:** 2025-02-13

**Authors:** Wenzhong Wang, Dongrui Zhang, Pia Heltoft Thomsen, Meng Sun, Ying Chang

**Affiliations:** 1Institute of Industrial Crops, Heilongjiang Academy of Agricultural Sciences, Harbin 150086, China; wenwen0331@163.com; 2College of Life Sciences, Northeast Agricultural University, Harbin 150030, China; dongruizhang96@gmail.com (D.Z.); mengsun1999@163.com (M.S.); 3Norwegian Institute of Bioeconomy Research, 1431 Ås, Norway; pia.heltoft@nibio.no

**Keywords:** *Fusarium oxysporum*, potato, dryocrassin ABBA, molecular mechanisms

## Abstract

Potato *Fusarium* dry rot and wilt are the most important soil- and seed-borne diseases in potatoes. They cause high economic losses during potato growth and storage across the world. Previous observations have shown that dryocrassin ABBA can induce resistance in potatoes. However, little is known about whether dryocrassin ABBA can suppress *Fusarium oxysporum*. In this research, we determined that exogenous dryocrassin ABBA significantly inhibited the mycelial growth, changed the cell ultrastructure, increased the MDA content, and decreased the antioxidant enzyme activity of *F. oxysporum*. The transcriptome analysis of *F. oxysporum* with or without dryocrassin ABBA indicated that 1244 differentially expressed genes (DEGs) were identified, of which 594 were upregulated and 650 were downregulated. GO term analysis showed that the DEGs were mostly related to biological processes. The KEGG pathway was mainly related to carbohydrate, amino acid, and lipid metabolism. Moreover, most of the expressions of PCWDEs, HSPs, and MFS were downregulated, decreasing the stress capacity and weakening the pathogenicity of *F. oxysporum* with dryocrassin ABBA treatment. These findings contribute to a new understanding of the direct functions of dryocrassin ABBA on *F. oxysporum* and provide a potential ecofriendly biocontrol approach for potato *Fusarium* dry rot and wilt.

## 1. Introduction

Potato (*Solanum tuberosum*) is a significant non-cereal food for human beings. Approximately 375 million tons of potatoes were produced globally in 2022 worldwide [1]. Potato *Fusarium* dry rot and wilt are important soil- and seed-borne diseases [2,3], which have been found in many potato-producing areas. They can lead to wilt symptoms, resulting in plant death in the field. They can also cause tuber rot during storage. Potato tuber yield loss ranges from 6% to 25% during storage, reaching up to 60–88% [4,5]. According to reports, potato *Fusarium* dry rot and wilt are caused by many species of *Fusarium*, including *F. oxysporum*, *F. solani*, and *F. culmorum* [2].

To manage *F. oxysporum* in agriculture, the first step is to plant resistant potato varieties; however, there are no varieties resistant to all *Fusarium* species. The second step comprises cultural practices, such as long rotations, the minimization of handling damage, and the control of storage conditions. At the moment, chemical fungicides are the most important and useful control method, including thiabendazole (TBZ), carbendazim, prochloraz, and iprodione; however, *Fusarium* isolates are becoming drug-resistant [2]. In order to prevent the development of resistance, many farmers use mixtures of fungicides. This can increase costs and cause fungicide residues and environmental pollution. Therefore, there is an exigent need to find some new alternatives to solve this problem. Biological management offers a sustainable, cost-effective, and environmentally friendly alternative approach to control *F. oxysporum*. Bioactive metabolites, which are chemical compounds produced by living organisms during their metabolic processes, have significant potential as novel chemical compounds [6]. Metacycloprodigiosin is a metabolite from the *Streptomyces alboflavus* KRO3 strain, which has demonstrated dose-dependent inhibitory activity against *F. oxysporum* f. sp. *pisi* [7]. Iturin A is an active antifungal compound of *Bacillus amyloliquefaciens* NCPSJ7 that causes the inhibition of *F. oxysporum* f. sp. *niveum* via dysfunction and an increase in membrane permeability [8]. A monoterpene phenol thymol (2-isopropyl-5-methylphenol) significantly reduced the pathogenicity and viability of *F. oxysporum* spores [9]. All these indicate that bioactive metabolites are a potential way of controlling plant pathogens, particularly *F. oxysporum.*

*Dryopteris crassirhizoma* has been used in traditional medicine in East Asia to treat epidemic flu [10], parasitic infestations [11], and cancer [12,13]. Dryocrassin ABBA, extracted from *D. crassirhizoma*, is a traditional Chinese medicine monomer and belongs to phloroglucinol. It has recently been reported to have inhibitory activity against viruses and bacteria; for example, dryocrassin ABBA has shown inhibitory activity against the main protease of SARS-CoV-2 [14,15] and against *Streptococcus pneumoniae* through a bactericidal effect and neutralizing pneumolysin activity [16]. All this research suggests that dryocrassin ABBA plays an important role in the prevention and treatment of many human diseases. It can be considered one of the most promising alternatives to synthetic chemical fungicides, which have negative effects on the environment and human health.

However, the antifungal activity of dryocrassin ABBA remains unknown. Our previous research indicated that *D. crassirhizoma* extracts showed good inhibition against *Fusarium* spp., including *F. solani* var. *coeruleum*, *F. culmorum*, *F. avenaceum*, and *F. sambucinum* [17,18]. Dryocrassin ABBA induced resistance to potato primarily through the salicylic acid (SA) signal pathway [19]. However, there is little information about whether or not dryocrassin ABBA can prevent and control *F. oxysporum*. The main aim of this experiment was to study the direct effects of dryocrassin ABBA on *F. oxysporum* via morphology. Mycelium intracellular reactive oxygen species (ROS) were analyzed via measurement of the malondialdehyde (MDA) content and four antioxidant enzymes’ activity. The molecular mechanisms were revealed via transcriptome analysis.

In this study, we conducted the following work: (1) identification of the direct effect of dryocrassin ABBA on mycelial growth and the change in the cell ultrastructure of *F. oxysporum* in vitro; (2) measurement of the MDA content and activities of four antioxidant enzymes of *F. oxysporum* treated with or without dryocrassin ABBA; and (3) exploration of the antifungal molecular mechanism of dryocrassin ABBA on *F. oxysporum* through transcriptome analysis.

## 2. Results

### 2.1. The Growth of F. oxysporum Can Be Significantly Inhibited by Dryocrassin ABBA

*F. oxysporum* was inoculated in a PDA medium with or without dryocrassin ABBA. The results showed that dryocrassin ABBA had an inhibitory effect on *F. oxysporum* compared with the control (water). The higher the dryocrassin ABBA concentration, the smaller the colony diameter (Figure 1a). When the dryocrassin ABBA concentration reached 2 g/L, the inhibition rate reached 93.13%, and the inhibition rate increased with the concentration (Figure 1b). The SEM results showed that there were deformed hyphae of *F. oxysporum* caused by the treatment of dryocrassin ABBA (Figure 1d), while the hyphae of the control sample were normal (Figure 1c). After treatment, the hyphae were twisted, and there were many blasted inflated mycelia. However, we did not observe the tightly twisted mycelium mentioned in the previous reports [17]. All this indicates that dryocrassin ABBA is able to inhibit the mycelial growth of *F. oxysporum*, with a significant positive correlation between them.

### 2.2. Variation in the Malondialdehyde Content During Dryocrassin ABBA Inhibition

After treatment, the content of MDA was higher in *F. oxysporum* mycelium than in the control group, i.e., it significantly increased by 1.83 times (Figure 2a). The results showed that the dryocrassin ABBA treatments caused membrane lipid peroxidation, increased the MDA content, and damaged the cell membrane.

### 2.3. Variation in the Antioxidant Enzyme Activity During Dryocrassin ABBA Inhibition

Antioxidant enzymes include superoxide dismutase (SOD), catalase (CAT), peroxidase (POD), and glutathione reductase (GR). After treatment, the CAT, POD, SOD, and GR activity of *F. oxysporum* mycelium significantly decreased compared with the control group (Figure 2b–e). All these results indicated that dryocrassin ABBA could inhibit *F. oxysporum* mycelium growth.

### 2.4. Analysis of Transcriptome Data

To further explore the role of dryocrassin ABBA in the inhibition of *F. oxysporum*, we performed transcriptome sequencing of *F. oxysporum* with or without 2.0 g/L of dryocrassin ABBA for 7 d. Clean data (clean reads) were obtained by removing the reads containing an adapter, reads containing more than 10% ploy-N, reads with all A bases, and low-quality reads from the raw data. After the screening, about 36,452,059 and 41,426,630 clean reads were obtained for the dryocrassin ABBA-treated sample and the control, respectively (Table 1). In the clean reads of the dryocrassin ABBA-treated sample, approximately 86.15% of the reads were mapped to the annotated reads of *F. oxysporum* “http://fungi.ensembl.org/Fusarium_oxysporum/Info/Index (accessed on 01 November 2022)”, of which approximately 80.55% and 5.59% were mapped to unique reads and multiple reads, respectively. In addition, approximately 91.13%, 83.54%, and 7.58% of the reads were mapped to the reads, unique reads, and multiple reads in the CK clean reads, respectively (Table 1).

In contrast to the samples that were not treated (CK), 1244 differentially expressed genes (DEGs) were identified in the dryocrassin ABBA-treated samples, of which 594 genes were upregulated and 650 genes were downregulated (Figure 3a). The GO terms and KEGG pathways were analyzed in order to further detect the inhibition mechanism of dryocrassin ABBA against *F. oxysporum* (Figure 3b,c). In the significantly enriched GO items (sorted by *p*-value), the top five terms were purine-containing compound metabolic process, nucleotide metabolic process, nucleoside monophosphate metabolic process, ATP metabolic process, and cation transmembrane transport (Appendix A), indicating that dryocrassin ABBA mainly inhibited biological processes in *F. oxysporum* and caused changes in the cell component. Among the top 20 enriched KEGG pathways, 17 significantly enriched KEGG pathways were related to metabolism, mainly related to carbohydrate, amino acid, and lipid metabolism (Figure 3c). In addition to metabolic pathways, dryocrassin ABBA also affected various processes associated with genetic information processing, cell transport and catabolism, and membrane transport (Appendix A).

To validate the Illumina RNA-Seq results, 10 DEGs were randomly chosen for quantitative real-time PCR (qRT-PCR). All 10 DEGs showed a similar expression trend in both qRT-PCR and RNA-seq, suggesting the reliability of the RNA-seq data (Figure 4).

### 2.5. DEGs Involved in F. oxysporum Cell Growth

In our data, most of the DEGs induced by the dryocrassin ABBA treatment were related to amino acid, carbohydrate, energy, and lipid metabolism, while the downregulated DEGs were predominant in metabolic pathways. Therefore, dryocrassin ABBA can inhibit mycelial growth by affecting lipid, amino acid, and carbohydrate metabolism (Figure 5). Amino acid and lipid metabolism are more important since they are effective pesticide targets [20]; thus, using dryocrassin ABBA to control potato dry rot or wilt might be an effective biological control method (Figure 5).

### 2.6. DEGs Related to the Plant-Cell-Wall-Degrading Enzymes (PCWDEs) of F. oxysporum

The basic components of the plant cell wall are cellulose, hemicellulose, pectin, lignin, and structural proteins [21]. When *F. oxysporum* overcomes the first barrier of the plant cell wall, some PCWDEs are activated and play significant roles [22,23]. This could facilitate mycelial spread and plant vessel blockage and cause the plant to wilt and die. The cellulose means cellulases hydrolyze the soluble cellodextrin oligomers to glucose. Cellulose-degrading enzymes are a group of complex enzymes, which include endo-β-1,4-glucanases, β-glucosidase, endoglucanase, and other enzymes. The transcriptome analysis showed that 11 DEGs were annotated as cellulose-degrading enzymes, and 9 were downregulated. Hemicellulose comprises xyloglucans, xylans, and galactomannans. Hemicellulose-degrading enzymes are a class of enzymes that break down hemicellulose, including mannanase, xylanases, galactanase, glucanase, etc. [24]. One DEG, beta-1,3-mannanase, was downregulated. Comprising different structures of the main chain of pectin, pectin-degrading enzymes encompass a variety of enzymes, including polygalacturonases (PGs), pectin lyases (PMGL), pectinesterase (PE), pectate lyases, etc. [25]. Pectin-degrading enzymes are one of the most important toxic factors for *Fusarium* spp. In our data, we found one DEG encoding pectin methylesterase and one encoding endopolygalacturonase; both were downregulated. Pathogenic fungi contain many CAZymes, such as cutinases and xylan esterases. In our data, we found one DEG encoding cutinase, which is a serine esterase, degraded fatty acids, triglycerides, etc. In the transcriptome data, we found that the expression of most PCWDEs in *F. oxysporum* was negatively affected (Table 2).

### 2.7. DEGs Related to Stress

The expression levels of some DEGs related to stress were also affected (Table 3). Heat shock proteins (HSPs) are common and conserved protein families. They are molecular chaperones that stabilize client proteins involved in various cell functions in fungi under different environmental stress conditions. HSPs are classified into different families such as HSP90, HSP70, HSP60, and small HSPs (sHSPs) [26]. In this study, three differentially expressed HSPs were downregulated, and 12 differentially expressed MFS transporters were mostly downregulated.

## 3. Discussion

About 70–80% of the losses in agricultural production are caused by fungal diseases [27], and over 8000 fungal species are known to cause around 100,000 different diseases in plants [28]. This has led to the increased use of fungicides. It has been reported that only 10% of fungicides reach the target organism, while 90% remain in the environment [29]. This has resulted in environmental pollution, thereby limiting fungicide use in agriculture across the world. Bioactive fungicides are considered one of the most promising alternatives because they are generally regarded as safe and environmentally friendly. Bioactive fungicides include microorganism and plant fungicides. Microorganism fungicides are produced under certain culture conditions, which have antagonistic activities and bioactive secondary metabolites against fungi. The inhibition effect may result from various environmental factors and can be difficult to control. Plant fungicides are produced by extraction. The bioactive components of plant fungicides are secondary metabolites, including terpenes and terpenoids, alkaloids, and phenolic compounds, which are easily absorbed by plants, with no residual hazards, and they do not change the soil and water environment. The mechanisms against phytopathogenic fungi are direct, such as the inhibition of hyphae growth, spore germination, protein and DNA synthesis, destruction of the cell wall, and membrane disruption [30]. Potato tubers or stems infected by *F. oxysporum* can cause yield and quality reduction. Moreover, *F. oxysporum* also produces mycotoxins that are potentially dangerous to human health. Some kinds of microorganisms and plant fungicides have been reported as bioactive fungicides to control *F. oxysporum*, such as *B. amyloliquefaciens*, *S. alboflavus*, thymol, chitosan, and osthole [7,8,9,20,31]. Dryocrassin ABBA is a phloroglucinol derivative, which is a kind of secondary metabolite synthesized by *D. crassirhizoma*. It belongs to plant fungicides, it is easy to produce and use, and it is not affected by environmental conditions, unlike microorganism fungicides. Compared with the essential oil of plant fungicides, dryocrassin ABBA is powdery and easy to transport and dissolve. In our study, dryocrassin ABBA treatment affected the growth of *F. oxysporum,* and the inhibition effect was dose-dependent (Figure 1). Therefore, dryocrassin ABBA, as a plant-derived reagent, greatly strengthens the potential to control soilborne and postharvest disease.

MDA is the final product of membrane lipid peroxidation [32]. Its content is an important indicator of the degree of membrane lipid peroxidation and is directly related to the degree of damage to the cell membrane [33]; the greater the degree of damage to the cell membrane, the higher the malondialdehyde content. In response to increased ROS levels and membrane lipid peroxidation, cells will use their own antioxidant enzymes to remove intracellular ROS to maintain oxidative equilibrium. POD eliminates hydrogen peroxide and phenol, amine, and aldehyde toxicity. SOD is the main antioxidant enzyme in mitochondria that can be cleared by O_2_^−^. GR is required for the resistance to oxidative stress because the deletion of the glutathione reductase gene commonly results in fungal mutants that are sensitive to various stressors [34]. Hydrogen peroxide (H_2_O_2_) is toxic to cells, and CAT dismutase can break down H_2_O_2_ into water and dioxygen [35]. In our study, dryocrassin ABBA increased the MDA content but decreased the activities of POD, SOD, CAT, and GR in *F. oxysporum* mycelia. This suggests that dryocrassin ABBA increases the MDA and H_2_O_2_ contents of the mycelia and induces oxidative stress in *F. oxysporum*, which may result in lipid peroxidation and cell leakage. This has a large effect on the synthesis and transport of sugars, proteins, and other substances, which results in slow growth and abnormal physiological metabolism [36].

Transcriptome analysis could provide insight into the mechanism, revealing a variety of biological processes involved in the inhibitory activity, such as melatonin against potato late blight caused by *Phytophthora infestans* [37] and chitosan and thymol against potato dry rot caused by *F. oxysporum* [20,36]. In our research, the transcriptomic analysis showed that there were 1244 DEGs. GO enrichment was associated with the inhibition of many biological processes of *F. oxysporum* and caused changes in the cell component. KEGG pathways were enriched in terms of metabolism, mainly related to carbohydrate, amino acid, and lipid metabolism, which indicated dryocrassin ABBA mainly inhibited many biological processes of *F. oxysporum,* causing changes in the cell component.

Lipid metabolism takes part in fungi germination and reproductive processes, and fatty acids store and provide energy [38]. Amino acid metabolism and carbohydrate metabolism provide proteins and energy. In addition, amino sugar, nucleotide sugar, and carbon metabolism pathways are involved in the growth and pathogenesis of *Fusarium* wilt in banana [39]. In this research, DEGs were predominant in metabolic pathways related to amino acid, carbohydrate, energy, and lipid metabolism and were downregulated.

The plant cell wall is an outer rigid semi-elastic supportive and protective layer. The composition and structure of the plant cell wall can differ significantly [21]. When *F. oxysporum* overcomes the first barrier of the plant cell wall, some PCWDEs are activated and play significant roles [22,23]. They could facilitate mycelial spread and plant vessel blockage and cause the plant to wilt and die. There was a correlation between the PCWDEs and the pathogenicity of the fungi [23]. In our data, nine DEGs annotated for cellulose-degrading enzymes, one annotated for beta-1,3-mannanase, one annotated for pectin methylesterase, one annotated for endopolygalacturonase, and one annotated for encoding cutinase were all downregulated (Table 2). Therefore, we believe dryocrassin ABBA can regulate the expression of PCWDE genes and weaken the pathogenicity of *F. oxysporum*.

HSPs have a strong effect on virulence and host–pathogen interactions. They are upregulated in response to various stresses, such as oxidative stress, nutrient deprivation, and antifungal drugs [40]. As reported, *Aspergillus fumigatus* modulated the expression of virulence factors by the upregulation of HSPs [41]. After treatment with chitosan, HSP expression in *F. oxysporum* showed a trend of downregulation [20].

Major facilitator superfamily (MFS) transporters were reported to take part in multidrug resistance in fungi. The overexpression of MFS can make cells resistant to drugs [42], and downregulated MFS genes in *P. infestans* can degrade pesticides [43]. MFS transporters play an important role in multidrug resistance in fungi [44], and many MFS transporters have been demonstrated to confer resistance to toxins and fungicides. In this study, three differentially expressed HSPs were downregulated, and twelve differentially expressed MFS transporters were mostly downregulated (Table 3). Our study indicates that dryocrassin ABBA weakens the stress capacity and multidrug resistance of *F. oxysporum* after dryocrassin ABBA treatment.

## 4. Materials and Methods

### 4.1. Materials

*F. oxysporum* was obtained from the laboratory of the Institute of Industrial Crops of Heilongjiang Academy of Agriculture Sciences (HAAS), Harbin, China. *F. oxysporum* was maintained on potato dextrose agar (PDA) at 25 °C in the dark.

Dryocrassin ABBA (5 mg, HPLC ≥ 98%) was purchased from Shanghai Yuanye Bio-Technology Co., Ltd. (Shanghai, China).

### 4.2. Measurements of the Effects of Dryocrassin ABBA on Mycelial Growth

The effects of dryocrassin ABBA on the mycelial growth of *F. oxysporum* were measured according to Yao and Tian [45]. The mycelial disks (6 mm in diameter) from 1-week-old fungal cultures were placed in the center of the Petri dishes (90 mm in diameter) with 10 mL of PDA containing different concentrations of dryocrassin ABBA at 0.5, 1, and 2 g/L and then incubated at 25 °C for 7 days in the dark. A treatment with distilled water was used as a control. The mycelial growth was determined by measuring the colony diameter on the PDA plate 7 days after inoculation. Each treatment was replicated 3 times, and the experiment was repeated twice. The mycelial growth inhibition of the treatment against the control was calculated using the following equation:

The mycelial growth inhibition = [(the mycelial growth of control − the mean mycelial growth of treatment)/the mycelial growth of control] × 100%.

### 4.3. Scanning Electron Microscopy (SEM)

The effect of dryocrassin ABBA on the hyphae of *F. oxysporum* was observed via SEM. The fungal cultures were treated with dryocrassin ABBA at 2 g/L in PDA and then incubated at 25 °C. The fungal culture treated with distilled water was used as a control. The samples were vapor-fixed with 2% (*w*/*v*) aqueous osmium tetroxide for 2 h at 4 °C, air-dried, and sputter-coated with gold palladium in a Polaron E 500 sputter coater (Polaron, Cambridge, UK). They were then kept in a desiccator until examination with a Cambridge Stereoscan 5–150 SEM (LEO Electron Microscopy Ltd., Cambridge, UK) operating at 20 kV. Micrographs were taken by a CCD-Camera (Gatan, Inc., Pleasanton, CA, USA). The experiment was repeated 3 times on 2 replicate plates for each treatment. For each replicate, 10 agar blocks were examined using scanning electron microscopy.

### 4.4. Enzyme Activity Assay

The fungal cultures were treated with dryocrassin ABBA at 2 g/L in PDA and then incubated for 7 d at 25 °C. The fungal culture treated with distilled water was used as a control. The activities of superoxide dismutase (SOD), catalase (CAT), peroxidase (POD), and glutathione reductase (GR) enzymes and the malondialdehyde (MDA) content were measured using a detection kit. The manufacture was Suzhou Comin Biotechnology Co., Ltd. (Suzhou, China). The detection kits number were SOD-2-W, CAT-1-W, POD-1-Y, GR-1-W and MDA-1-Y. All the measurements were conducted according to the detection kit manuals.

### 4.5. Transcriptome Assay and the Dataset Analysis

*F. oxysporum* was cultured in a PDA medium with or without 2 g/L of dryocrassin ABBA at 25 °C in the dark for 7 days. The mycelia were collected and kept at −80 °C for transcriptome studies. Total RNA was isolated using a TRIzol total RNA extraction kit (TIANGEN, Cat. No. DP424), which yielded > 2 μg of total RNA per sample. All the samples were sent to APExBIO (Shanghai, China) for transcriptome sequencing. The library was sequenced using the Illumina NovaSeq 6000 sequencing platform (Illumina, San Diego, CA, USA) (Paired end150) to generate raw reads.

Oligo-dT primers were used to transverse mRNA to obtain cDNA (APExBIO, Cat. No. K1159). Reference genome and gene model annotation files were downloaded from the genome website directly “http://fungi.ensembl.org/Fusarium_oxysporum/Info/Index (accessed on 1 November 2022)”. An index of the reference genome was built using Hisat2 v2.0.5, and paired-end clean reads were aligned to the reference genome using Hisat2 v2.0.5. Feature Counts v1.5.0-p3 was used to count the number of reads mapped to each gene. Then, the FPKM of each gene was calculated based on the length of the gene and the read count mapped to that gene. PA *p*-value *<* 0.05 and absolute foldchange greater than 2 were set as the thresholds for significant differential expression. Gene Ontology (GO) and the Kyoto Encyclopedia of Genes and Genomes (KEGG) enrichment analysis of differentially expressed genes were implemented using the cluster Profiler R package (version 4.15.1). GO terms and KEGG pathways with corrected *p*-values *<* 0.05 were considered significantly enriched.

### 4.6. RNA Extraction and Real-Time PCR

All the primers were synthesized commercially by BSGENE (Harbin, China). The gene ID and gene-specific primers are listed in Appendix A. The extraction method of the total RNA of *F. oxysporum* was the same as the transcriptome analysis. The reaction was achieved using an RT-PCR kit. The manufacture was JiangSu CoWin Biotech Co., Ltd. (Taizhou, China). The detection kit number was CW0957. The reaction was conducted in a Roche Light Cycler 480 Real-Time PCR cycler. The manufacture was F. Hoffmann-La Roche & Co., Basel, Switzerland. Each experiment was repeated three times, and the relative amounts of the amplification products were analyzed using the 2^−ΔΔCt^ method [46].

## 5. Conclusions

In our research, we found that dryocrassin ABBA significantly inhibited mycelial growth, increased MDA content, and decreased the CAT, POD, SOD, and GR activities of *F. oxysporum*. Based on the analysis of the transcriptome data, we found that dryocrassin ABBA affected lipid, amino acid, and carbohydrate metabolism. Most expressions of PCWDEs, HSPs, and MFS were downregulated, the stress capacity was decreased, and the pathogenicity of *F. oxysporum* was weakened by dryocrassin ABBA treatment. This research mainly described the effect of dryocrassin ABBA on *F. oxysporum* and verified the mechanism of dryocrassin ABBA against *F. oxysporum* at the expression level. Based on our findings and previous studies [17,19], dryocrassin ABBA could be used against potato dry rot, including the direct inhibition of *Fusarium* spp. to induce resistance in potatoes. These findings suggest that dryocrassin ABBA might be a promising natural fungicide for the potato dry rot and wilt caused by *F. oxysporum*.

## Figures and Tables

**Figure 1 ijms-26-01573-f001:**
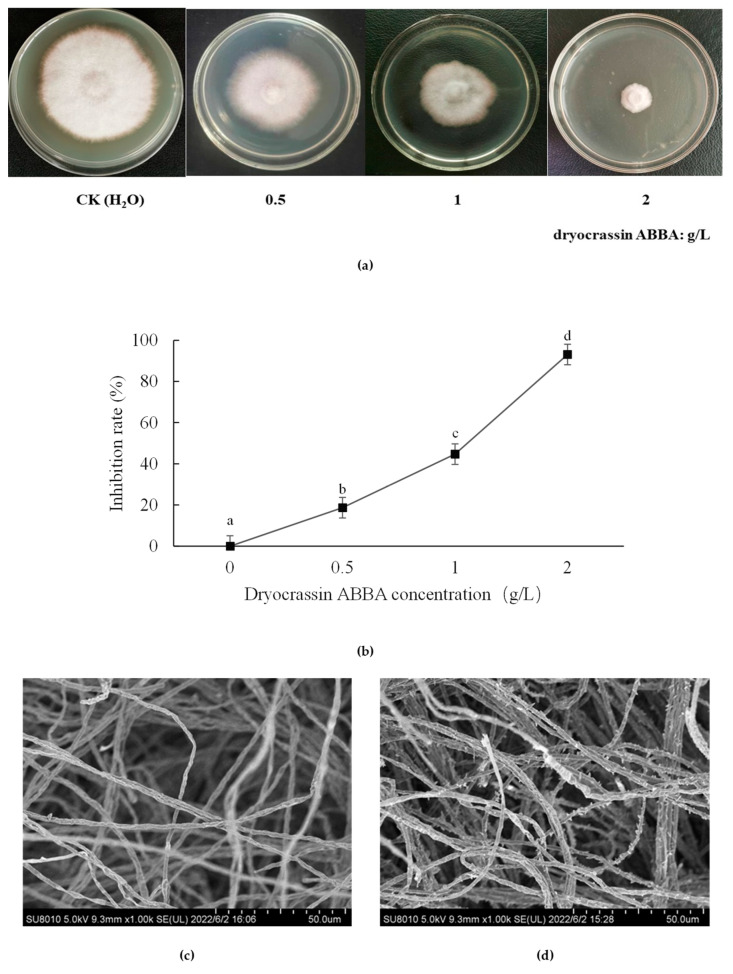
Dryocrassin ABBA’s inhibition effect on *F. oxysporum* growth. (**a**) Mycelial phenotypes of *F. oxysporum* grown on a PDA medium at different concentrations for 7 d, with distilled water as the control. (**b**) Rate of dryocrassin ABBA inhibition on the pathogenic ability of *F. oxysporum* (different lowercase letters indicate *p* < 0.01). (**c**) The morphology of *F. oxysporum* hyphae treated with water, as observed using scanning electron microscopy (SEM). (**d**) The morphology of *F. oxysporum* hyphae treated with dryocrassin ABBA, as observed using SEM.

**Figure 2 ijms-26-01573-f002:**
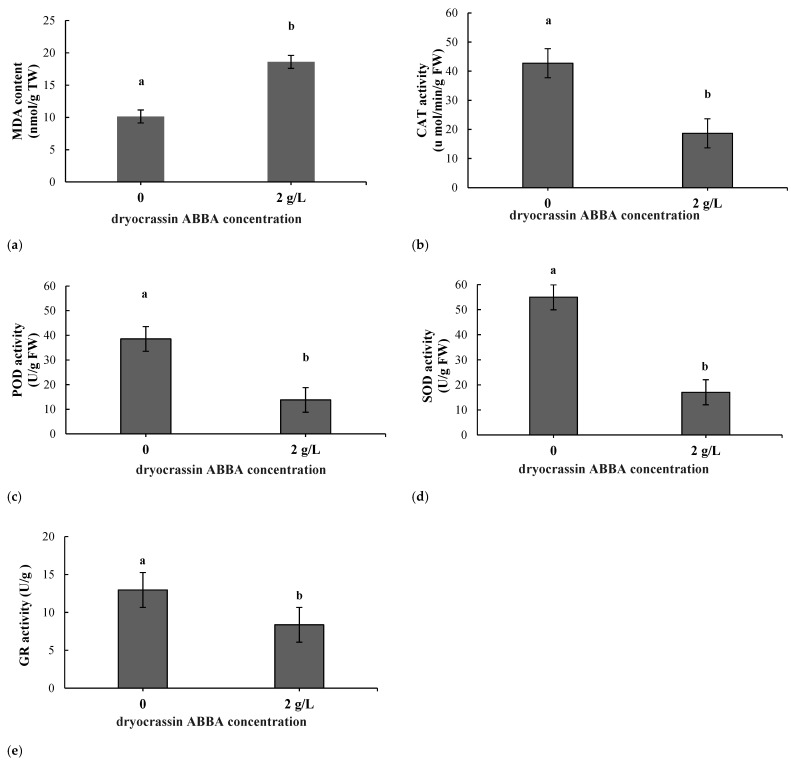
The results of the malondialdehyde content and antioxidant enzyme activity of *F. oxysporum mycelium* after dryocrassin ABBA treatment. (**a**) Malondialdehyde (MDA) content of *F. oxysporum* mycelium. (**b**) Catalase (CAT) content of *F. oxysporum* mycelium. (**c**) Peroxidase (POD) activity of *F. oxysporum* mycelium. (**d**) Superoxide dismutase (SOD) activity of *F. oxysporum* mycelium. (**e**) Glutathione reductase (GR) activity of *F. oxysporum* mycelium after dryocrassin ABBA treatment. (Different lowercase letters indicate *p* < 0.01).

**Figure 3 ijms-26-01573-f003:**
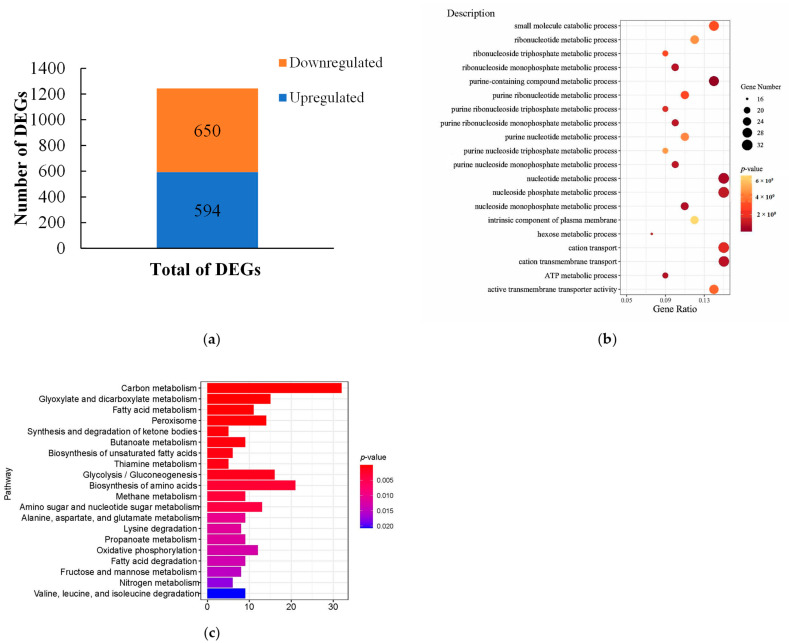
Analysis of the transcriptome data of *F. oxysporum* treated with or without dryocrassin ABBA. (**a**) Number of significant up/downregulated genes. (**b**) Scatter diagram of enriched GO terms. (**c**) Top 20 enriched KEGG pathways.

**Figure 4 ijms-26-01573-f004:**
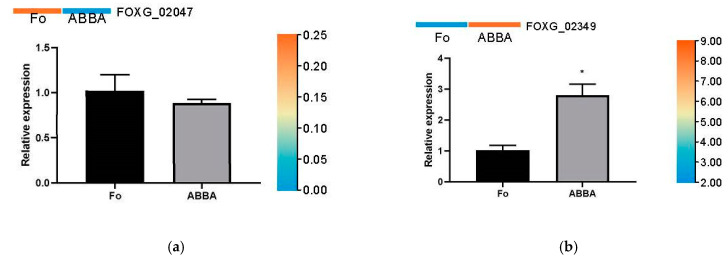
Validation results of the Illumina RNA-Seq, relative expression of 10 DEGs without or with the treatment of dryocrassin ABBA (**a**–**j**). Note: The error bars represent the standard deviation of the mean relative expression value. The left *Y*-axis represents the relative expression level of different expression genes, and the right *Y*-axis represents the FPKM value of transcriptome sequencing. The heatmap at the top of the bar chart represents the expression of Illumina RNA-Seq results. Significant differences from the control are denoted as asterisks (*) at *p* < 0.05.

**Figure 5 ijms-26-01573-f005:**
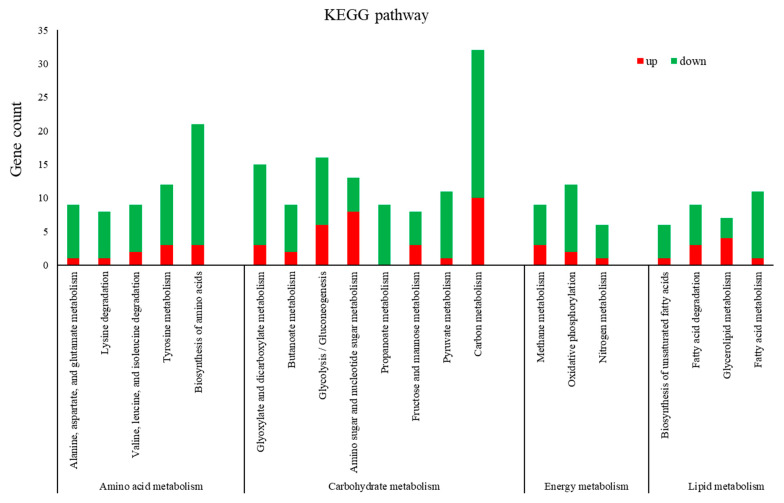
Dryocrassin ABBA can inhibit some KEGG pathways involved in lipid, amino acid, and carbohydrate metabolism in *F. oxysporum*.

**Table 1 ijms-26-01573-t001:** Summary statistics for *F. oxysporum* treated without or with dryocrassin ABBA (T) based on the RNA-Seq data.

	Not Treated (CK)	Dryocrassin ABBA-Treated (T)
Clean reads	39,858,919	36,452,059
Total reads	41,426,630	37,971,115
Clean base	5.87 Gb	5.32 Gb
GC content	50.60%	51.24%
% ≥ Q30	92.23%	92.70%
Mapped reads	36,312,056 (91.13%)	31,334,678 (86.15%)
Unique mapped reads	33,308,578 (83.54%)	29,254,757 (80.55%)
Multiple map reads	3,003,478 (7.58%)	2,079,921 (5.59%)

**Table 2 ijms-26-01573-t002:** DEGs related to plant-cell-wall-degrading enzymes (PCWDEs).

	Gene Id	log2FoldChange	*p*-Value	Product	Stat
Pectin	FOXG_12330	−2.94633	0.000616	hypothetical protein similar to pectin methylesterase [Source: BROAD_F_oxysporum; Acc: FOXG_12330]	down
	FOXG_13051	−3.77406	9.03 × 10^−8^	endopolygalacturonase [Source: UniProtKB/TrEMBL; Acc: Q14TQ5]	down
Hemicellulase	FOXG_13531	−2.10907	6.49 × 10^−6^	hypothetical protein similar to beta-1,3-mannanase [Source: BROAD_F_oxysporum; Acc: FOXG_13531]	down
	FOXG_15742	3.349943	4.94 × 10^−12^	endo-1,4-beta-xylanase [Source: UniProtKB/TrEMBL; Acc: Q9C1R1]	up
Cellulose	FOXG_02349	1.859403	2.19 × 10^−7^	beta-glucosidase 1 precursor [Source: BROAD_F_oxysporum; Acc: FOXG_02349]	up
	FOXG_08942	−2.58522	0.001399	alpha-glucosidase [Source: BROAD_F_oxysporum; Acc: FOXG_08942]	down
	FOXG_09571	−1.18895	0.001978	hypothetical protein similar to beta-glucosidase [Source: BROAD_F_oxysporum; Acc: FOXG_09571]	down
	FOXG_16943	−3.43939	3.66 × 10^−19^	hypothetical protein similar to glucosidase [Source: BROAD_F_oxysporum; Acc: FOXG_16943]	down
	FOXG_02921	−3.40605	0.000175	hypothetical protein similar to beta-1,6-glucanase precursor [Source: BROAD_F_oxysporum; Acc: FOXG_02921]	down
	FOXG_07831	−1.39445	9.04 × 10^−6^	hypothetical protein similar to endo-1,3(4)-beta-glucanase [Source: BROAD_F_oxysporum; Acc: FOXG_07831]	down
	FOXG_10638	−2.92964	6.24 × 10^−5^	putative endoglucanase type K [Source: UniProtKB/Swiss-Prot; Acc: P45699]	down
	FOXG_02349	1.859403	2.19 × 10^−7^	beta-glucosidase 1 precursor [Source: BROAD_F_oxysporum; Acc: FOXG_02349]	up
	FOXG_08942	−2.58522	0.001399	alpha-glucosidase [Source: BROAD_F_oxysporum; Acc: FOXG_08942]	down
	FOXG_09571	−1.18895	0.001978	hypothetical protein similar to beta-glucosidase [Source: BROAD_F_oxysporum; Acc: FOXG_09571]	down
	FOXG_16943	−3.43939	3.66 × 10^−19^	hypothetical protein similar to glucosidase [Source: BROAD_F_oxysporum; Acc: FOXG_16943]	down
Hemicellulase	FOXG_13531	−2.10907	6.49 × 10^−6^	hypothetical protein similar to beta-1,3-mannanase [Source: BROAD_F_oxysporum; Acc: FOXG_13531]	down
	FOXG_15742	3.349943	4.94 × 10^−12^	endo-1,4-beta-xylanase [Source: UniProtKB/TrEMBL; Acc: Q9C1R1]	up

**Table 3 ijms-26-01573-t003:** DEGs related to stress.

Gene id	log2FoldChange	*p*-Value	Product	Stat
FOXG_00233	−6.557428176	4.57 × 10^−13^	30 kDa heat shock protein [Source: BROAD_F_oxysporum; Acc: FOXG_00233]	down
FOXG_00378	−2.622230433	0.000330041	heat shock protein 78, mitochondrial precursor [Source: BROAD_F_oxysporum; Acc: FOXG_00378]	down
FOXG_09418	−3.508647595	2.78 × 10^−5^	heat shock protein HSP98 [Source: BROAD_F_oxysporum; Acc: FOXG_09418]	down
FOXG_09887	−2.381667113	0.000485036	hypothetical protein similar to major facilitator superfamily protein superfamily [Source: BROAD_F_oxysporum; Acc: FOXG_09887]	down
FOXG_00029	1.47992372	4.79 × 10^−10^	hypothetical protein similar to MFS amino acid transporter [Source: BROAD_F_oxysporum; Acc: FOXG_00029]	up
FOXG_00479	1.033004122	0.004485202	hypothetical protein similar to MFS quinate transporter QutD [Source: BROAD_F_oxysporum; Acc: FOXG_00479]	up
FOXG_02047	−6.856468946	0.000331397	hypothetical protein similar to mfs multidrug-resistance transporter [Source: BROAD_F_oxysporum; Acc: FOXG_02047]	down
FOXG_03950	−2.749470704	1.33 × 10^−11^	hypothetical protein similar to MFS toxin efflux pump [Source: BROAD_F_oxysporum; Acc: FOXG_03950]	down
FOXG_04943	9.588338475	6.41 × 10^−11^	hypothetical protein similar to MFS myo-inositol transporter [Source: BROAD_F_oxysporum; Acc: FOXG_04943]	up
FOXG_09269	−2.999258742	3.04 × 10^−6^	hypothetical protein similar to MFS drug transporter [Source: BROAD_F_oxysporum; Acc: FOXG_09269]	down
FOXG_09811	−2.960490486	2.44 × 10^−5^	hypothetical protein similar to MFS peptide transporter [Source: BROAD_F_oxysporum; Acc: FOXG_09811]	down
FOXG_12240	−5.672136552	1.73 × 10^−17^	hypothetical protein similar to mfs multidrug-resistance transporter [Source: BROAD_F_oxysporum; Acc: FOXG_12240]	down
FOXG_15236	−7.14892256	3.85 × 10^−17^	hypothetical protein similar to mfs multidrug-resistance transporter [Source: BROAD_F_oxysporum; Acc: FOXG_15236]	down
FOXG_15681	2.796322541	2.58 × 10^−5^	hypothetical protein similar to MFS peptide transporter [Source: BROAD_F_oxysporum; Acc: FOXG_15681]	up
FOXG_17534	5.910328445	0.001304151	hypothetical protein similar to MFS sugar transporter [Source: BROAD_F_oxysporum; Acc: FOXG_17534]	up

## Data Availability

Data are contained within the article and Appendix A.

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
