# Peer review of "Inhibitory Effect and Mechanism of Dryocrassin ABBA Against Fusarium oxysporum"

_ijms, 2025, doi:10.3390/ijms26041573_

Round 1
Reviewer 1 Report
Comments and Suggestions for Authors
In this study, the authors explore the inhibitory effect of dryocrassin ABBA on the fungal growth of Fusarium oxysporum and perform transcriptomic analysis to investigate the underlying mechanisms. The manuscript is well-written and presents some interesting findings. However, there are several concerns need to be addressed:
1. While the study demonstrates a 93.13% inhibition rate at a concentration of 2 g/L, this concentration seems relatively high and may not be practical for the management of potato dry rot and wilt diseases caused by F. oxysporum.
2. Previous studies have shown that ABBA exerts its effects by targeting specific proteins. Could the authors please rationalize why it's appropriate by using transcriptomic analysis to explore the mechanism of ABBA's action.
3. Several studies related to the mechanism of ABBA’s function were not cited in the manuscript.
4. The Figures need to be reorganized.
Author Response
|
Comments 1: While the study demonstrates a 93.13% inhibition rate at a concentration of 2 g/L, this concentration seems relatively high and may not be practical for the management of potato dry rot and wilt diseases caused by F. oxysporum. |
|
Comments 2: Previous studies have shown that ABBA exerts its effects by targeting specific proteins. Could the authors please rationalize why it's appropriate by using transcriptomic analysis to explore the mechanism of ABBA's action |
|
Response 2: Agree. The previous studies have shown that ABBA exerts its effects by targeting specific proteins. In this research we used transcriptomic analysis to explain the mechanism of ABBA on F. oxysporum. Because the researches about inhibition against Fusarium spp and some other plant pathogens were analyzed by transcriptome. These will easy to compare molecular mechanisms and find it effect the same different expressions, pathway and so on.
Comments 3 Several studies related to the mechanism of ABBA’s function were not cited in the manuscript. Response 3: Agree. I have, accordingly, modified and add the inhibition mechanism of ABBA’s function to emphasize this point. In the revised manuscript this change can be found – page 12, and line 216-244.]
Comments 4 The Figures need to be reorganized. Response 4: Agree. I have, accordingly modified all the Figures to emphasize this point.
|
Reviewer 2 Report
Comments and Suggestions for Authors
I have reviwed the manuscript from Wenzhong Wang et al. which addresses a potential interesting research to the field. However, it requires major revisions to meet the standards for suggesting it for publication. Please, consider my revisions, detailed below:
- The manuscript must undergo professional English revision. I noticed numerous instances of grammatically incorrect sentences and inaccurate vocabulary throughout the text. Avoid vague or imprecise expressions such as “there were a lot of” (line 68). Instead, use quantitative or specific descriptions to maintain precision.
- While the topic is well summarized in the introduction, this section is currently too brief. I encourage you to expand the introduction to include the following key topics:
a) Discuss current methods for managing Fusarium oxysporum in agriculture. For example, are fungicides the primary option? If so, which ones are commonly used? Are these methods sufficient? Highlight their specific limitations.
b) Provide a brief overview of recent research into bioactive metabolites with inhibitory activity against F. oxysporum, which is a topic covered by diverse recent articles that have described potential for metabolites such as metacycloprodigiosin (https://doi.org/10.3390/jof10110783), iturin A (https://doi.org/10.3390/foods11192996), or thymol (https://doi.org/10.1016/j.postharvbio.2022.112025), among others. This contextualization will demonstrate progress in this area and highlight the significance of the manuscript and of the searching for bioactive metabolites in combating F. oxysporum.
c) If available, please provide an overview of additional biological activities described for dryocrassin ABBA, such as its phytotoxicity, antifungal properties against other fungi, antibacterial activity, etc.
- The size of the figures and their accompanying text must be increased. Currently, they are too small and difficult to read. For example, Figures 1a, 1b, 2, 3, and 5 all need resizing for improved legibility.
- Ensure that the footnotes accompanying figures are complete and self-explanatory. For instance, the footnote for Figure 4 is currently incomplete. The footnotes should thoroughly describe all aspects of the figure, including statistical treatments, explanations of abbreviations, and any other relevant details.
- While the discussion is comprehensive, I suggest adding some lines to evaluate how dryocrassin ABBA compares to existing bioactive fungicide molecules. Where possible, draw comparisons based on parameters such as availability, ease of production, physicochemical properties, alternative modes of action, and lower toxicity. Use existing literature and your research findings to support these comparisons.
Comments on the Quality of English LanguageThe English language requires thorough revision.
Author Response
|
Comments 1: The manuscript must undergo professional English revision. I noticed numerous instances of grammatically incorrect sentences and inaccurate vocabulary throughout the text. Avoid vague or imprecise expressions such as “there were a lot of” (line 68). Instead, use quantitative or specific descriptions to maintain precision. |
|
Response 1: Thank you for pointing this out. I agree with this comment. The manuscript has edited by professional English revision.
|
|
Comments 2: While the topic is well summarized in the introduction, this section is currently too brief. I encourage you to expand the introduction to include the following key topics: a) Discuss current methods for managing Fusarium oxysporum in agriculture. For example, are fungicides the primary option? If so, which ones are commonly used? Are these methods sufficient? Highlight their specific limitations. b) Provide a brief overview of recent research into bioactive metabolites with inhibitory activity against F. oxysporum, which is a topic covered by diverse recent articles that have described potential for metabolites such as metacycloprodigiosin (https://doi.org/10.3390/jof10110783), iturin A (https://doi.org/10.3390/foods11192996), or thymol (https://doi.org/10.1016/j.postharvbio.2022.112025), among others. This contextualization will demonstrate progress in this area and highlight the significance of the manuscript and of the searching for bioactive metabolites in combating F. oxysporum. c) If available, please provide an overview of additional biological activities described for dryocrassin ABBA, such as its phytotoxicity, antifungal properties against other fungi, antibacterial activity, etc. |
|
Response 2: Agree. The introduction is too brief, I expanded the introduction with current methods for managing F. oxysporum in agriculture. I added the recent articles inhibitory activity against F. oxysporum. dryocrassin ABBA and additional biological activities.
Comments 3 The size of the figures and their accompanying text must be increased. Currently, they are too small and difficult to read. For example, Figures 1a, 1b, 2, 3, and 5 all need resizing for improved legibility. Response 3: Agree. I have increased the figures and their accompanying text.
Comments 4 Ensure that the footnotes accompanying figures are complete and self-explanatory. For instance, the footnote for Figure 4 is currently incomplete. The footnotes should thoroughly describe all aspects of the figure, including statistical treatments, explanations of abbreviations, and any other relevant details. Response 4: Agree. I have accordingly modified the footnote for Figure 4.
Comments 5: While the discussion is comprehensive, I suggest adding some lines to evaluate how dryocrassin ABBA compares to existing bioactive fungicide molecules. Where possible, draw comparisons based on parameters such as availability, ease of production, physicochemical properties, alternative modes of action, and lower toxicity. Use existing literature and your research findings to support these comparisons Response 5: Agree. I have modified discussion section. Make the comparisons between dryocrassin ABBA and other bioactive fungicide. In the revised manuscript this change can be found – page 12, and line 62-71.]
|
Round 2
Reviewer 2 Report
Comments and Suggestions for Authors
Authors have improved the manuscript, adressing all the suggestions I gave.
- Just in lines 50-51, please correct "Metacycloprodigiosin is an extract" for "Metacycloprodigiosin is a metabolite".
- The quality of the text in Figure 2 should be improved.
Comments on the Quality of English LanguageIt is correct
Author Response
|
Comments 1: Just in lines 50-51, please correct "Metacycloprodigiosin is an extract" for "Metacycloprodigiosin is a metabolite". Response 1: Thank you for pointing this out. I agree with this comment. Therefore, I have revised manuscript this change can be found – page2, and line 50-51. |
|
Comments 2: The quality of the text in Figure 2 should be improved. Response 2: Agree. I have modified Figure 2 to emphasize this point. In the revised manuscript this change can be found – page 4, and line 117-118. |